# Age Structure, Development and Population Viability of Banteng (*Bos javanicus*) in Captive Breeding for Ex-Situ Conservation and Reintroduction

**DOI:** 10.3390/ani13020198

**Published:** 2023-01-05

**Authors:** Rattanawat Chaiyarat, Neeracha Sriphonkrang, Phattaranan Khamsirinan, Saree Nakbun, Namphung Youngpoy

**Affiliations:** 1Wildlife and Plant Research Center, Faculty of Environment and Resource Studies, Mahidol University, Nakhon Pathom 73170, Thailand; 2Khao Nampu Nature and Wildlife Education Center, Department of National Parks, Wildlife and Plant Conservation, Kanchanaburi 71250, Thailand

**Keywords:** banteng, captive breeding, population viability analysis, reintroduction program, Salakphra Wildlife Sanctuary

## Abstract

**Simple Summary:**

Captive breeding is important for ex-situ conservation and future reintroduction. This research aims to conduct a population viability analysis (PVA) for a sustainable reintroduction program for bantengs. The monthly development of 23 founder individuals was assessed. The PVA showed that the time required to reach the maximum population in a captive banteng program is dependent on the carrying capacity of the habitat. The reduction of a small banteng founder group by the reintroduction of animals into the wild can negatively affect the population growth of the captive group. This information can be used to maintain the population viability of bantengs and sustain ex-situ conservation and the reintroduction program in Thailand and elsewhere.

**Abstract:**

Captive breeding is important for ex-situ conservation and the future reintroduction of bovids that become extinct in the wild. The age structure, development, and viability of captive-bred bantengs (*Bos javanicus*) are important to sustain the long-term reintroduction program in Salakphra Wildlife Sanctuary (SWF) and other areas. This research conducted a long-term population viability analysis (PVA) using height, weight, body condition scores (BSC), age structure, and development in captivity for a sustainable reintroduction program of bantengs in Thailand. Monthly development photographs of 23 founder individuals (12 males and 11 females) were assessed by three banteng experts, two researchers, and three members of the general public. The assessments of weight and BCS were not significantly different among the three groups, while height was underestimated by the general public. The PVA showed that the time to reach the maximum population in a captive banteng program is dependent on the carrying capacity of the habitat. The reduction of a small banteng founder group by the reintroduction of animals into the wild can negatively affect the population growth of the captive group. This information can be used to maintain the population viability of bantengs and sustain ex-situ conservation and the reintroduction program in Thailand and elsewhere.

## 1. Introduction

Habitat fragmentation, poaching, illegal logging, and mining have reduced the global and national population of the banteng (*Bos javanicus*) [1]. The construction of roads and agricultural areas, as well as poaching, are the biggest threats to the banteng in Thailand [2]. Globally, bantengs are listed as Endangered by The International Union for Conservation of Nature (IUCN) [3] and listed as Critically Endangered in Thailand [4]. The global population is approximately 5000–8000 individuals [5]. Bantengs are found in parts of southeast Asia, such as Myanmar, Thailand, Indochina, and Java [6]. In Thailand, there were 470 individuals in 1995 [7]. Recently, the populations have been increasing in many areas, although the exact numbers are not known. Bantengs were extirpated in the Salakphra Wildlife Sanctuary when it was first established as a nature preserve. After more than 30 years of local extinction, a reintroduction program under the guideline of IUCN/SSC [8] was established between 2014 and 2022. Since then, 16 captively bred bantengs have been reintroduced into the wild [9].

Bantengs prefer open dry deciduous forests and avoid evergreen rainforests but occupy secondary forest formations that are a result of logging and fires and enter tracts of sub-humid forest on occasion in the more humid areas of Java and Borneo (Wharton, 1968). However, their predominant habitat type is the tropical lowland dipterocarp forest in Sabah, Malaysia [10], at elevations lower than 900 m [11]. In Thailand, the highest density was found in Huai Kha Kheng Wildlife Sanctuary. Sometimes they have been found feeding in agricultural areas such as cassava and coconut plantations [12].

In Thailand, bantengs feed on 150 species of food plants [13], with 23 of these species found in Khao Khiao-Khao Chomphu Wildlife Sanctuary [12] and 24 species in Salakphra Wildlife Sanctuary [14]. They are often found near saltlicks and water sources [13]. Bantengs can be found living together with other mammals, such as sambar (*Rusa unicolor*) and wild boar (*Sus scrofa*) [15]. Bantengs are highly active in the late afternoon and at dusk. They prefer resting under shrubs, and seasonal changes do not affect their activities [16]. They forage in groups containing between 2 and 30 individuals. Herds mostly comprise adult and subadult females, with adult males joining the herd during mating season [2]. Solitary animals tend to be mature bulls or sometimes old cows. The composition of small groups of cows with calves or juveniles and the solitary state of old individuals may remain the same for months or even years.

The bantengs found in this study were collected from diverse areas and kept at the Khao Nam Phu Nature and Wildlife Education Center (KNP). A female banteng calf named Pongtong was found and captured by local people at the Pongtong saltlick located in Salakphra Wildlife Sanctuary. This calf was sent to the Special Warfare Command, Lopburi Province, before being sent to KNP for breeding. One male was found in the forest area of Kamphaeng Phet Province. Two other males came from Kaeng Krachan National Park and Huai Kha Khaeng Wildlife Sanctuary (S.N.; head of KNP, personal communication) [17]. The life history of each animal at KNP, such as name, sex, date of birth, identifying characteristics, vaccination, and pedigree, were recorded along with monthly photographs. This information was used to select suitable bantengs for reintroduction to the Salakphra Wildlife Sanctuary [9,14,18].

The objective of this study was to determine the age structure, growth (weight and height), and body condition score to develop banteng body metrics and assessment. We also projected the number of captive individuals and their age structure into the future to assess the ability of the captive population to indicate the continued viability of the captive-bred banteng population and the ability of the program to yield animals for reintroduction to ex-situ populations in Thailand and elsewhere.

## 2. Materials and Methods

### 2.1. Life History of Bantengs in Khao Nam Phu Nature and Wildlife Education Center

The ages of 23 bantengs (12 males and 11 females) kept in KNP were identified from photographs and studbook data between 2007–2019 (Table 1).

### 2.2. Relationship among Age, Height, Weight, and Body Condition Score of Captive Banteng

The photographs of each individual banteng were used to assess age (year), weight (kg), height (m) and BSC monthly. The height and weight were described using morphology as for gaur [19]. Only sets of photos in which all parts of the body were visible were selected for analysis. Each individual was identified by obvious morphological characteristics such as scars, the shape of horns, and collars [18]. The weights and heights of the captive bantengs were assessed by comparison with a pole of known height in the enclosure. Three groups of assessors (specialists, researchers and untrained assessors) recorded the individual heights and weights monthly. The results in height and weight estimated by the researchers were validated by comparing them with the specialists and untrained assessors. The correlation between the age, weight and height of the captive bantengs was determined using general linear models in an R package [20]. The results were validated with the Bornean banteng [5].

The condition of seven body parts was rated using a five-level scoring system [18] (Table 1). The Spearman’s rank correlation between categorical body composition and the BCS of 23 bantengs was calculated using an R package [20] to assess changes in their condition [21]. Chi-square was used to compare the BCS of the bantengs between categorical factors such as wet and dry seasons. A Mann–Whitney–Wilcoxon rank of the BCS categories of the banteng was determined using SPSS [22]. The differences between treatments were tested at a *p* < 0.05 significance level.

### 2.3. Prediction of Population Number of Captive Banteng

We used Vortex v10 [23], parameterized with a combination of expert opinion and empirical data. The PVA was performed using various pieces of information to maximize the reliability of the model based on the IUCN/SSC Asian Wild Cattle Specialist Group (AWCSG), Gardner et al. [5] and KNP studbook (Table 2). The captive population used for population viability modelling in this study originally had 23 individuals and was reduced to 16 individuals when seven bantengs were reintroduced into SWF. The population models were designed to extend for 100 years (approximately 10 generations). The correlation between reproduction and survival as a function of environmental variance was 0.2 and was expected to be relatively low due to the K-selected nature of the species (i.e., low years for reproduction are not always low years for survival). The reproductive system was categorized as short-term polygyny [5]. The ages of females and males of the first offspring were 3 and 4 years, respectively (KNP studbook). The maximum age for reproductive females and males was 15 years (KNP studbook). The maximum lifespan was 26 years (KNP studbook), and the max number of calving events per year was 1 calf per year (KNP studbook). The sex ratio at birth was 50:50 [5]. The intrinsic model of percentage breeding adult females per year (interbirth interval) was 50% (KNP studbook). The percentage mortality of females was 20 ± 10% at age 0–1 year, 2 ± 5% at 1–2 years, 4 ± 5% at 2–3 years and 11.9 ± 2% at >3 years, with a maximum lifespan of ~28 years. The percentage mortality of males was 26 ± 10% at age 0–1 year, 8 ± 5% at 1–2 years, 11.9 ± 2% at 2–3 years and 11.9 ± 2% at >3 years, with a maximum lifespan of ~28 years (AWCSG). The intrinsic model of males in the breeding pool showed the highest potential at 100% (KNP studbook). The initial population size was 23 or 16 individuals, and the initial carrying capacity was 50 individuals (KNP studbook). All population modelling was carried out using Vortex v10 [23]. A key piece of literature used to guide this process was Lacy et al. [24]. Sensitivity testing based on the percentage of females breeding per year was carried out using parameters from Table 1 and 1000 simulations.

## 3. Results

### 3.1. Growth Rate of Captive Banteng

The development of bantengs was different between males and females; this is called sexual dimorphism. The morphology of male bantengs differed among age classes, especially after three years old, and especially their horn curvature, body size, skin color and dewlap. The male banteng’s color is generally dark brown, and its body is larger than the female’s. The male’s height may extend up to 160 cm, and its weight from 600–800 kg. The morphology of the female banteng only changes slightly after three years old. The females are thinner and usually pale brown or chestnut red. The female’s height may extend up to 140 cm, with a weight of 590–670 kg (Figure 1).

### 3.2. Relationship among Age, Height, Weight and Body Condition Score of Captive Banteng

The three groups of assessors evaluated the relationship between age and height of captive bantengs with the same trend as logistic growth into three phases: (1) 1–4 years was highly increased; (2) 4–6 years was slowly increased, and (3) 6–12 years was a steady state increase. The height assessment of the specialists was higher than researchers and untrained assessors (Table 3, Figure 2, Appendix A).

The three groups of assessors evaluated the relationship between the age and weight of captive bantengs. The assessment between professionals and researchers was not significantly different (*p* > 0.05), while the untrained assessors provided different results (*p* < 0.001) (Table 3, Figure 3, and Appendix A).

The body condition scores of bantengs increased with age, as evaluated by the three assessor groups (Table 3, Figure 4 and Appendix A).

### 3.3. Prediction of Population Number of Captive Banteng

At present, the captive breeding facility of KNP has 30 individuals, but if KNP wants to increase and sustain captive breeding capacity at 50, 60, 80 and 100 individuals in the next 100 years, what is the possible hypothetical carrying capacity that it should be? The prediction for the banteng population in captivity in the next 100 years, if the carrying capacity estimates by KNP were 30, 50, 60, 80 and 100 individuals and assuming there is no removal of individuals from the captive population during this population growth, was predicted. PVA estimated a banteng founder with 23 individuals would initially reach these populations in 2, 8, 4, 13 and 6 years, respectively (Figure 5a), while a starting population of 16 bantengs will reach the carrying capacity in 5, 6, 9, 11 and 13 years, respectively (Figure 5b). These models did not include unexpected stochastic events such as disease outbreaks, natural disasters, etc.

## 4. Discussion

### 4.1. Growth Rate of Captive Banteng

There were differences between the development of male and female bantengs. The morphology of male bantengs differed in horn shape, body size, skin color and dewlap at all ages, especially after three years old [2], while the morphology of female bantengs had only slight differences after three years old, as reported by Copland [25].

### 4.2. Relationship among Age, Height, Weight, and Body Condition Score of Captive Banteng

The height assessment of specialists was higher than researchers and untrained assessors due to the specialists being more familiar with the banteng than the researchers and the untrained group. It has been previously noted that variations in assessments may reduce the ability to assess the characteristics of the banteng [26]. However, relative changes in height assessed by all three groups in this study were similar.

The assessments of body condition by the three assessor groups were similar. The body condition score of bantengs increased with age in both males and females, as found in a previous study of reintroduced bantengs in Salakphra Wildlife Sanctuary [18]. This finding indicated that the food and nutrition supplied were suitable even though it was lower than the diet available in Salakphra Wildlife Sanctuary [14]. The males had higher BCS than females as male bantengs used a high BCS for courtship, while the BCS of females was reduced when they were lactating [18]. This study did not note any diseases or other factors affecting BCS as reported in wild bantengs in Indonesia [27,28,29].

### 4.3. Prediction of Population Number of Captive Banteng

Starting with the founder group of 23 individuals, if the KNP expands its carrying capacity to 30, 50, 60, 80, and 100 individuals, the banteng will reach these numbers in 2, 8, 4, 13 and 6 years, respectively. However, when the number of founders is reduced to 16 due to reintroductions, reaching carrying capacity is expected to be delayed to 5, 6, 9, 11 and 13 years, respectively. A small captive founder group is affected more by reductions in size through the reintroduction program. A small founder group may also lead to inbreeding. The severity of inbreeding depression varies widely among species [30,31]. To reach this possible hypothetical, KNP needs to prepare for captive breeding capacity, such as increasing the carrying capacity of captivity, increasing staff, increasing the quality and quantity of forages, and finding long-term funding support. The limitation of these models was to assume there is no removal of individuals from the captive population during this population growth and lack of genetic information of KNP. For future works, these models should include unexpected stochastic events, such as disease outbreaks, natural disasters, genetic considerations, etc., as the parameters.

### 4.4. Conservation and Management

If a captive banteng population is lower than ~20 individuals, increasing the population to an expected stable population for 100 years may take longer than a group of higher than 20 individuals. This possible hypothetical will be successful if the Department of National Parks, Wildlife and Plant Conservation (DNP) agrees, facilitate, and supports the KNP to continue the sustainable banteng captive breeding program. In general speculation, a captive breeding program with approximately 20 founder individuals can result in inbreeding depression and a decline in the captive population after 40 years if no individuals are translocated to the wild [5]. However, starting with a small group can be an optimal interim strategy if it allows time to establish recruitment mechanisms [32]. Breeding technology such as artificial insemination (AI) and embryo transfer [33,34,35] may be appropriate to increase the breeding banteng population. A translocation strategy of wild individuals to increase the gene pool of the captive program in KNP is important to increase the population genetics and recovery of the population. This activity will be a more effective use of biological and financial resources to prevent the extinction of the reintroduced banteng in Salakphra Wildlife Sanctuary and other areas.

## 5. Conclusions

Therefore, the correct management for breeding pairs was to monitor growth and body metrics to raise healthy captive individuals and predict the population viability of captive breeding bantengs to manage the suitable number of captive bantengs. For future studies, their genetic diversity is integral to minimizing inbreeding depression within a captive setting. Low genetic variability is currently an issue for captive-bred bantengs in Thailand [31] and other wildlife species [36,37], such as those found in the Malayan gaur (*Bos gaurus hubbacki*) where only a few founder individuals resulted in multiple progenies with shared parents. This affects the survival rate of newborn calves [38,39]. More than one ex-situ breeding facility supplied with multiple wild-caught individuals originating from different management units would be preferable. This methodology can not only minimize mortality arising from disease transmission but also maintain wildlife security and avoid unintentional reintroduction, such as when a captive-bred banteng escapes from its enclosure [12].

## Figures and Tables

**Figure 1 animals-13-00198-f001:**
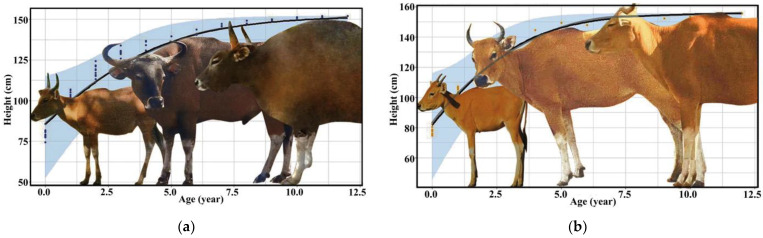
Development of morphology and height of (**a**) male and (**b**) female banteng in captivity.

**Figure 2 animals-13-00198-f002:**
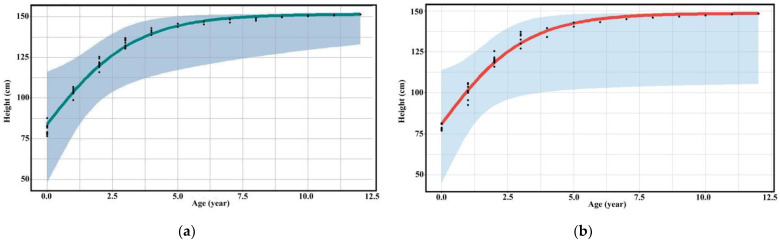
Scatter diagram (black dots) with regression line of the relationship between age and height of (**a**) male and (**b**) female captive banteng by researchers.

**Figure 3 animals-13-00198-f003:**
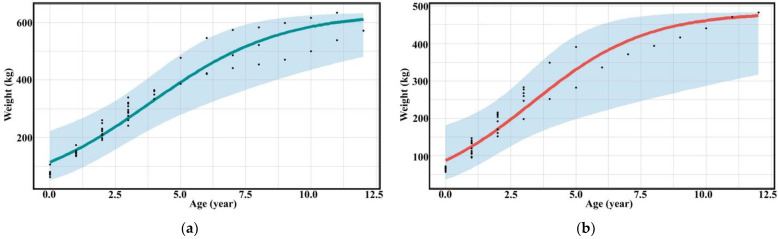
Scatter diagram (black dots) with regression line of the relationship between age and weight of (**a**) male and (**b**) female captive banteng by researchers.

**Figure 4 animals-13-00198-f004:**
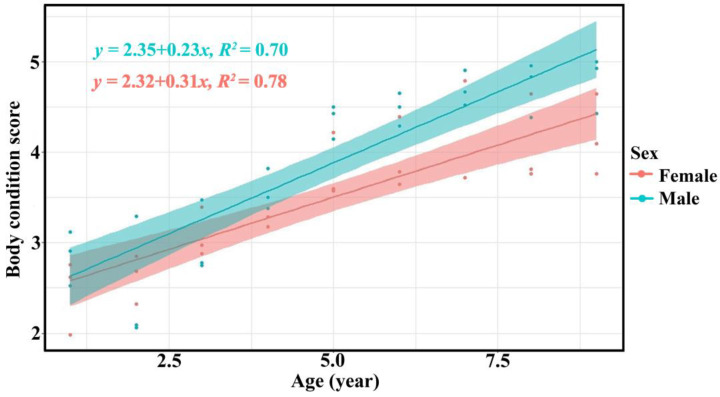
Relationship between age and body condition score of male (green) and female (red) captive bantengs by researchers.

**Figure 5 animals-13-00198-f005:**
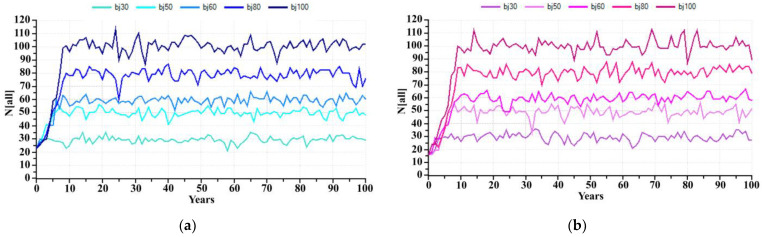
The number of captive banteng in 100 years with the carrying capacity at 30, 50, 60, 80 and 100 individuals; (**a**) 23 founders and (**b**) 16 founders.

**Table 1 animals-13-00198-t001:** The scoring criteria for body composition of the bantengs in captivity.

Body Composition ^1^	Body Condition Scoring System (BCS)				
	1	2	3	4	5
Neck	Very flat	Starting to round appearance	Slightly rounded	Rounded	Full and rounded
Dewlap	Non-existent	Present but very small	Present	Present with fat apparent	Full of fat, a large flap of skin
Shoulder	The shoulder bones are prominent	Covered with soft tissue in some part	Almost covered with soft tissue	Well-covered with soft tissue	The shoulders and forequarters are very well covered with soft tissue
Vertebrae	Prominent all along backbone	Easily seen	Visible with some tissue covered	Covered with soft tissue, but shadows can be seen	Not visible and covered with soft tissue
Ribs	Very prominent	Easily seen but covered by some tissue	Can be seen but covered by soft tissue	Rarely visible	Not visible
Hook	The top is flat, while pins and hooks may be prominent, hollow or dished	The top is flat with obvious hooks and pins	The top is flat or slightly rounded, but hooks are still visible	The top is flat or slightly rounded	The top is rounded, the hook may be slightly visible but flushed with tissue, and hind legs below are very full
Tail head	Very flat	Flat	Slightly rounded	Rounded	Rounded and has small mounds of soft tissue

^1^ Applied from Kongsurakan et al. [18].

**Table 2 animals-13-00198-t002:** Population parameters for bantengs in Khao Nam Phu Nature and Wildlife Education Center.

Parameter ^1^	Model Value	Justification
Number of years modelled	100	Approximately 10 generations
Correlation between reproduction and survival as a function of environmental variance	0.2	Unknown but expected to be relatively low due to K-selected nature of the species (i.e., low years for reproduction are not always low years for survival)
Reproductive system	Short-term polygyny	Gardner et al. (2021)
Age (yr) of first offspring–females	3	Information from KNP studbook
Age (yr) of first offspring–males	4	Information from KNP studbook
Max age (yr) female reproduction	15	Information from KNP studbook
Max age (yr) male reproduction	15	Information from KNP studbook
Max lifespan (yr)	26	Information from KNP studbook
Max number of calving events per year	1	Information from KNP studbook
Max number of offspring per calving event	1	Can ignore very rare twinning events
Sex ratio at birth	50:50	No other data
Density dependence	none	Intrinsic model
% adult females breeding per year (interbirth interval)	50	Information from KNP studbook
% Female mortality ± SD	20 ± 10% (Age 0–1 yr)	
	2 ± 5% (Age 1–2 yrs)	
	4 ± 5% (Age 2–3 yrs)	
	11.9 ± 2% (Age 3+ yrs)	
	For subadults, AWCSG data from *B. j. javanicus*. For adults, estimated from the North American Region Banteng studbook (both sexes)-equates to a max lifespan of ~28 yrs	
% Male mortality ± SD	26 ± 10% (Age 0–1 yr)	
	8 ± 5% (Age 1–2 yrs)	

^1^ Applied from Gardner et al. (2021) [5] and studbook of the Khao Nam Phu Nature and Wildlife Education Center (KNP).

**Table 3 animals-13-00198-t003:** Relationship between age (*y*), height, weight and body condition score (BCS) of captive banteng.

Parameter	Sex	Model	R^2^	*p*-Value
Height	Both	*yh* = 151.5 + 0.23 × 0.26ln(*x*)	0.41	<0.001
Height	Male	*yh* = 151.50 + 0.28 × 0.26ln(*x*)	0.28	<0.001
Height	Female	*yh* = 148.50 + 0.19 × 0.26ln(*x*)	0.30	0.001
Weight	Both	*yw* = 634.24 – 2.27 × 0.18ln(*x*)	0.056	0.16
Weight	Male	*yw* = 634 – 2.16 × 0.19ln(*x*)	−0.06	0.236
Weight	Female	*yw* = 483 – 2.42 × 0.17ln(*x*)	−0.04	0.48
BCS	Both	*yb* = 1.80 + 0.58*x*	0.69	<0.001
BCS	Male	*yb* = 2.32 + 0.31*x*	0.78	<0.001
BCS	Female	*yb* = 2.35 + 0.23*x*	0.70	<0.001

The model indicates the relationship between age (*x*) and growth parameters (height (*yh*), weight (*yw*), and body condition score (*yb*), ln = natural logarithm) in males, females, and both sexes.

## Data Availability

Not applicable.

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
