# Peer review of "Age Structure, Development and Population Viability of Banteng (Bos javanicus) in Captive Breeding for Ex-Situ Conservation and Reintroduction"

_animals, 2023, doi:10.3390/ani13020198_

Round 1

Reviewer 1 Report

Major Comments

The authors have compiled a nice set of data, models and collaborators. To better express the relevance of this research, the authors should re-write the objectives of this paper. Specifically, the objectives should be clear, and explicitly linked with the methods and results.  As present, it is unclear why the body measurements are being collected, whether they are linked with the PVA in any way, and why several kinds of assessors are being compared.  It is unclear how this data collection is relevant to the study objective. The discussion should reflect on the stated objectives, and provide caveats to the interpretation of results and highlight future research and conservation needs.

 Specific Comments

English grammar review recommended throughout to insert missing words and correct singular/plural errors e.g., abstract 21-22, 75, 84, 121, 210

 49 – Extirpated instead of extinct

 87-91:  I find the objectives statement to be unclear. Please be more specific about the research objective addressed in this paper. 

How will you objectively interpret these measures (growth rate, age structure and condition score and population viability metrics) to assess the sustainability of the program, and the wild populations?

 121 – Suggest starting with a statement that you used Vortex, parameterized with a combination of expert opinion and empirical data.

 136 – Confusing. Suggest more clearly expressing mortality rates  as percentages e.g., , at age 0-1 year was 20 (+/- 10%).

 Reword to increase clarity: 140 (model of males was 100%)

 Cool graphics in Figure 1!

 161 – Here or methods, please add the reason why you used 3 groups of assessors and created different models of age-height/weight/condition. You present many alternative relationships in Table 3, Figures 2-7; but the reader does not really understand what you are trying to demonstrate. Why are these differences important? How do they relate to your study objective? Suggest moving most/all of these graphics to the supplement. Retain only the information deemed as reliable for use in the PVA or assessments that relate to your study goal. Explorations of (un)reliable data should reside in the supplement or be omitted.

 Table 3: Does sex refer to the cattle or the assessor? If the assessor, suggest omitting.

 209 -210:  Add explanation of ‘KNP’ i.e., captive breeding facility. Add a statement indicating if these carrying capacities are feasible/possible, or hypothetical.  These time to carrying capacity estimates assume there was no removal of individuals (i.e., releases) from the captive population during this phase of population growth. Please add these and other assumptions to the text. Please state whether unexpected stochastic events such as disease outbreaks are accounted for in the model parameterization.

 217-221 (Discussion) – This is phrased as a result, with no discussion. Please add a statement that links this finding to your study objectives. 

 It does not appear that any benteng growth or body condition metrics were used in the PVA, so this paper seems disjointed.  I suggest you add objectives to your introduction section that relate to developing benteng body metrics (and assessments) that indicate the ability of the captive breeding program to raise healthy captive individuals (or other specific purpose).

 Please add a section that discusses model caveats, and lists future information needs.

252 – Inbreeding after 40 years – is this a result of the vortex model or general speculation? Please indicate whether you included genetic considerations in the methods. Your conclusions emphasize the need to manage genetic diversity, but it is unclear whether you modeled genetics and no genetic results were presented. Suggest you more closely link your concluding statements to your study objectives.

Author Response

Dear Reviewer #1

Please find the Responded to Comments and Suggestions for Authors of Reviewer #1

Major Comments

The authors have compiled a nice set of data, models and collaborators. To better express the relevance of this research, the authors should re-write the objectives of this paper. Specifically, the objectives should be clear, and explicitly linked with the methods and results. As present, it is unclear why the body measurements are being collected, whether they are linked with the PVA in any way, and why several kinds of assessors are being compared. It is unclear how this data collection is relevant to the study objective. The discussion should reflect on the stated objectives, and provide caveats to the interpretation of results and highlight future research and conservation needs.

Responded: Thank you for your positive comments. We changed the objective as “The objective of this study was to determine the age structure, growth (weight and height), body condition score to developing banteng body metrics and assessment that indicate the ability of the captive breeding program to raise healthy captive individuals, and predicted population viability of captive breeding banteng for manage the suitable number of captive banteng for sustainability of the ex-situ conservation and reintroduction programs in Thailand and elsewhere” in line 80-85.

Specific Comments

English grammar review recommended throughout to insert missing words and correct singular/plural errors e.g., abstract 21-22, 75, 84, 121, 210

Responded: We recheck the English grammar throughout the manuscript.

49 – Extirpated instead of extinct

Responded: We changed “extinct” to “extirpated” as “Banteng were extirpated in the Salakphra Wildlife Sanctuary when it was first established as a nature preserve” in line 46-47.

87-91: I find the objectives statement to be unclear. Please be more specific about the research objective addressed in this paper.

Responded: We changed to “The objective of this study was to determine the age structure, growth (weight and height), body condition score to developing banteng body metrics and assessment that indicate the ability of the captive breeding program to raise healthy captive individuals, and predicted population viability of captive breeding banteng for manage the suitable number of captive banteng for sustainability of the ex-situ conservation and reintroduction programs in Thailand and elsewhere” in line 80-85.

How will you objectively interpret these measures (growth rate, age structure and condition score and population viability metrics) to assess the sustainability of the program, and the wild populations?

Responded: We changed to “The objective of this study was to determine the age structure, growth (weight and height), body condition score to developing banteng body metrics and assessment that indicate the ability of the captive breeding program to raise healthy captive individuals, and predicted population viability of captive breeding banteng for manage the suitable number of captive banteng for sustainability of the ex-situ conservation and reintroduction programs in Thailand and elsewhere” in line 80-85.

121 – Suggest starting with a statement that you used Vortex, parameterized with a combination of expert opinion and empirical data.

Responded: We added “We used Vortex v10 [23], parameterized with a combination of expert opinion and empirical data” in line 116-117.

136 – Confusing. Suggest more clearly expressing mortality rates as percentages e.g., , at age 0-1 year was 20 (+/- 10%).

Responded: We changed to “Mortality rates as percentage of females at age 0−1 year was 20±10%), 1−2 year was 2±5%, 2−3 year was 4±5% and > 3 years was 11.9±2% with a maximum lifespan of ~28 year. Percentage mortality of males at age 0−1 year was 26±10%, 1−2 year was 8±5%, 2−3 year was 11.9±2% and >3 years was 11.9±2% with a maximum lifespan of ~28 year (AWCSG)” in line 131-135.

Reword to increase clarity: 140 (model of males was 100%)

Responded: We changed to “The intrinsic model of percentage breeding adult females per year (interbirth interval) was 50% (KNP studbook)” in line 130-131.

Cool graphics in Figure 1!

Responded: Thank you.

161 – Here or methods, please add the reason why you used 3 groups of assessors and created different models of age-height/weight/condition.

Responded: We added the reason of used 3 groups of assessors as “Three groups of assessors (specialists, researchers and untrain assessors) recorded the individual heights and weights monthly. The results in height and weight estimate of researchers were validated by compared with specialists and untrain assessors” in line 101-104.

You present many alternative relationships in Table 3, Figures 2-7; but the reader does not really understand what you are trying to demonstrate. Why are these differences important?

Responded: These differences were important for compared the validity of height, weight and BSC of searchers when compared with specialists and untrain assessors.

How do they relate to your study objective?

Responded: These are not related to our objective, but we want to showed that the researcher must be trained before observe.

Suggest moving most/all of these graphics to the supplement. Retain only the information deemed as reliable for use in the PVA or assessments that relate to your study goal. Explorations of (un)reliable data should reside in the supplement or be omitted.

Responded: We move most of these graphics to the supplement, and retain only the information that reliable for used in the PVA.

Table 3: Does sex refer to the cattle or the assessor? If the assessor, suggest omitting.

Responded: We referred to cattle.

209 -210: Add explanation of ‘KNP’ i.e., captive breeding facility. Add a statement indicating if these carrying capacities are feasible/possible, or hypothetical. These time to carrying capacity estimates assume there was no removal of individuals (i.e., releases) from the captive population during this phase of population growth. Please add these and other assumptions to the text. Please state whether unexpected stochastic events such as disease outbreaks are accounted for in the model parameterization.

Responded: We added “At present, captive breeding facility of KNP was 30 individuals, but if KNP want to increase and sustain the captive breeding capacity at 50, 60, 80 and 100 individuals in the next 100 years, what is the possible hypothetical carrying capacity that it should be. The prediction of banteng population in captivity in the next 100 year, if the carrying capacity estimates of KNP was 30, 50, 60, 80 and 100 individuals and assume there is not removal of individuals from the captive population during this population growth, was predicted. PVA estimated a banteng founder with 23 individuals will initially reach these populations in 2, 8, 4, 13 and 6 years, respectively (Fig. 5a), while a starting population of 16 banteng will reach the carrying capacity in 5, 6, 9, 11 and 13 years, respectively (Fig. 5b). These models were not included unexpected stochastic events such as disease outbreaks, natural disasters, etc.” in line 189-199.

217-221 (Discussion) – This is phrased as a result, with no discussion. Please add a statement that links this finding to your study objectives.

Responded: We added “To reach this possible hypothetical, KNP need to prepare the captive breeding capacity such as increase the carrying capacity of captivity, increasing staff, increasing the quality and quantity of forages, and finding the long-term funding support. The limitation of these models was assume there is not removal of individuals from the captive population during this population growth and lack of genetic information of KNP. For future works, These models should include unexpected stochastic events such as disease outbreaks, natural disasters, and genetic considerations, etc. as the parameters” in line  229-236.

It does not appear that any benteng growth or body condition metrics were used in the PVA, so this paper seems disjointed. I suggest you add objectives to your introduction section that relate to developing benteng body metrics (and assessments) that indicate the ability of the captive breeding program to raise healthy captive individuals (or other specific purpose).

Responded: We added “The objective of this study was to determine the age structure, growth (weight and height), body condition score to developing banteng body metrics and assessment that indicate the ability of the captive breeding program to raise healthy captive individuals, and predicted population viability of captive breeding banteng for manage the suitable number of captive banteng for sustainability of the ex-situ conservation and reintroduction programs in Thailand and elsewhere” in line        .

Please add a section that discusses model caveats, and lists future information needs.

Responded: We added “To reach this possible hypothetical, KNP need to prepare the captive breeding capacity such as increase the carrying capacity of captivity, increasing staff, increasing the quality and quantity of forages, and finding the long-term funding support. The limitation of these models was assume there is not removal of individuals from the captive population during this population growth and lack of genetic information of KNP. For future works, These models should include unexpected stochastic events such as disease outbreaks, natural disasters, and genetic considerations, etc. as the parameters” in line  229-236.

252 – Inbreeding after 40 years – is this a result of the vortex model or general speculation? Please indicate whether you included genetic considerations in the methods. Your conclusions emphasize the need to manage genetic diversity, but it is unclear whether you modeled genetics and no genetic results were presented. Suggest you more closely link your concluding statements to your study objectives.

Responded: We added “Therefore, correct management of breeding pairs was to monitor growth and body metrics of to raise healthy captive individuals and predicted population viability of captive breeding banteng for manage the suitable number of captive banteng. The future study, their genetic diversity is integral to minimizing inbreeding depression within a captive setting” in line 253-257.

Responded: We added “At present, captive breeding facility of KNP was 30 individuals, but if KNP want to increase and sustain the captive breeding capacity at 50, 60, 80 and 100 individuals in the next 100 years, what is the possible hypothetical carrying capacity that it should be. The prediction of banteng population in captivity in the next 100 year, if the carrying capacity estimates of KNP was 30, 50, 60, 80 and 100 individuals and assume there is not removal of individuals from the captive population during this population growth, was predicted. PVA estimated a banteng founder with 23 individuals will initially reach these populations in 2, 8, 4, 13 and 6 years, respectively (Fig. 5a), while a starting population of 16 banteng will reach the carrying capacity in 5, 6, 9, 11 and 13 years, respectively (Fig. 5b). These models were not included unexpected stochastic events such as disease outbreaks, natural disasters, etc.” in line 189-199.

217-221 (Discussion) – This is phrased as a result, with no discussion. Please add a statement that links this finding to your study objectives.

Responded: We added “To reach this possible hypothetical, KNP need to prepare the captive breeding capacity such as increase the carrying capacity of captivity, increasing staff, increasing the quality and quantity of forages, and finding the long-term funding support. The limitation of these models was assume there is not removal of individuals from the captive population during this population growth and lack of genetic information of KNP. For future works, These models should include unexpected stochastic events such as disease outbreaks, natural disasters, and genetic considerations, etc. as the parameters” in line  229-236.

It does not appear that any benteng growth or body condition metrics were used in the PVA, so this paper seems disjointed. I suggest you add objectives to your introduction section that relate to developing benteng body metrics (and assessments) that indicate the ability of the captive breeding program to raise healthy captive individuals (or other specific purpose).

Responded: We added “The objective of this study was to determine the age structure, growth (weight and height), body condition score to developing banteng body metrics and assessment that indicate the ability of the captive breeding program to raise healthy captive individuals, and predicted population viability of captive breeding banteng for manage the suitable number of captive banteng for sustainability of the ex-situ conservation and reintroduction programs in Thailand and elsewhere” in line        .

Please add a section that discusses model caveats, and lists future information needs.

Responded: We added “To reach this possible hypothetical, KNP need to prepare the captive breeding capacity such as increase the carrying capacity of captivity, increasing staff, increasing the quality and quantity of forages, and finding the long-term funding support. The limitation of these models was assume there is not removal of individuals from the captive population during this population growth and lack of genetic information of KNP. For future works, These models should include unexpected stochastic events such as disease outbreaks, natural disasters, and genetic considerations, etc. as the parameters” in line  229-236.

252 – Inbreeding after 40 years – is this a result of the vortex model or general speculation? Please indicate whether you included genetic considerations in the methods. Your conclusions emphasize the need to manage genetic diversity, but it is unclear whether you modeled genetics and no genetic results were presented. Suggest you more closely link your concluding statements to your study objectives.

Responded: We added “Therefore, correct management of breeding pairs was to monitor growth and body metrics of to raise healthy captive individuals and predicted population viability of captive breeding banteng for manage the suitable number of captive banteng. The future study, their genetic diversity is integral to minimizing inbreeding depression within a captive setting” in line 253-257.

Reviewer 2 Report

The paper titled “Age Structure, Development and Population Viability of Banteng (Bos javanicus) in Captive Breeding for Ex-situ Conservation and  Reintroduction” attempted to establish the conservation  strategy for banteng.  The argument is interesting and however, the study is not  well written in style. Overall study seems to have novelty but require major changes in order to be considered for publishing in Animals.

A few shortcoming/suggestions are listed here:-

Line 162. The criteria and number of persons in  three groups of assessors needs to be elaborated in detail.

Line 169. It is not clear what Table 3 indicates. Please explain the details in the table footnote how values explain the relationship.

Line 203. “3.2. Figures, Tables and Schemes 203 All figures and tables should be cited in the main text as Figure 1, Table 1, etc”.  What is this ?. Please check and delete.

Line 270. The study was started with ex-situ methods and recommended in situ. Please recheck and clarify the statement.   

The study has many figures which may be reduced to minimum. Moreover, colored figures may be prepared to be sensible even if printed in black and white.

The connections and fluidity is missing in the article.

The objective and results hardly match.

The biological/physiological  explanation of results needs to be provided.

The use of technology for conservation was never discussed such as semen or embryo etc.

Author Response

Dear Reviewer #2

Please find the Responded to Comments and Suggestions for Authors of Reviewer #2

The paper titled “Age Structure, Development and Population Viability of Banteng (Bos javanicus) in Captive Breeding for Ex-situ Conservation and Reintroduction” attempted to establish the conservation strategy for banteng. The argument is interesting and however, the study is not well written in style. Overall study seems to have novelty but require major changes in order to be considered for publishing in Animals.

Responded: Thank you for your positive comments.

A few shortcoming/suggestions are listed here:-

Line 162. The criteria and number of persons in three groups of assessors needs to be elaborated in detail.

Responded: We added “Three groups of assessors (specialists, researchers and untrain assessors) recorded the individual heights and weights monthly. The results in height and weight estimate of researchers were validated by compared with specialists and untrain assessors” in line 101-104.

Line 169. It is not clear what Table 3 indicates. Please explain the details in the table footnote how values explain the relationship.

Responded: We added “The model indicates the relationship between age (y) and growth parameters (height, weight, and body condition score) in male, female, and both sexes” in the footnote of table 3.

Line 203. “3.2. Figures, Tables and Schemes 203 All figures and tables should be cited in the main text as Figure 1, Table 1, etc”. What is this ?. Please check and delete.

Responded: We recheck all of these.

Line 270. The study was started with ex-situ methods and recommended in situ. Please recheck and clarify the statement.

Responded: This affects the survival rate of newborn calves [38,39]. More than one ex-situ breeding facility supplied with multiple wild-caught individuals originating from different management units would be preferable” in line 260-262.

The study has many figures which may be reduced to minimum. Moreover, colored figures may be prepared to be sensible even if printed in black and white.

Responded: We move most of them to Supplementary Figure. For colored, we will reconstruct in the final manuscript.

The connections and fluidity is missing in the article.

Responded: We will rewrite the structure and English in the final article.

The objective and results hardly match.

Responded: We rewrote the objectives as “The objective of this study was to determine the age structure, growth (weight and height), body condition score to developing banteng body metrics and assessment that indicate the ability of the captive breeding program to raise healthy captive individuals, and predicted population viability of captive breeding banteng for manage the suitable number of captive banteng for sustainability of the ex-situ conservation and reintroduc-tion programs in Thailand and elsewhere” in line 80-85.

The biological/physiological explanation of results needs to be provided.

Responded: We added “The development of banteng was different between males and females or call sexual dimorphism. The morphology of male banteng differed among age classes, especially after three years old, and especially their horn curvature, body size, skin color and dewlap. The males color are generally dark brown and body is larger than female. The males height may up to 160 cm and weight at 600-800 kg. The morphology of female banteng only had slight changes after three years old. The females are thinner and usually pale brown or chestnut red. The females height may up to 140 cm with the weight at 590-670 kg (Figure 1)” in line 149-156.

The use of technology for conservation was never discussed such as semen or embryo etc.

Responded: We added “The breeding technology such as artificial insemination (AI) and embryo transfer [33-35] may appropriate to increase the breeding banteng population in line 245-247.

Best regards,

Round 2

Reviewer 1 Report

Thank you for addressing many of the last comments and suggestions. This version is improved.

The new objectives statement helps clarify the intentions of the paper, but is a really long run-on sentence. Please divide into 2 or more sentences: e.g., The objectives of this study were to determine the age structure, growth (weight and height), body condition score to developing banteng body metrics and assessments that indicate the ability of the captive breeding program to raise healthy captive individuals. We also projected the number of captive individuals and the age structure into the future to assess the ability of the captive population to indicate continued viability of captive-bred banteng population and the ability of the program to yield animals for reintroduction to ex-situ populations in Thailand and elsewhere.

English edit needed in some places. 

182: Rephrase: the assessment of the three assessor groups were similar, and move to after the second sentence, when the topic changes.

238: “If a banteng population is lower than ~20 individuals, a stable population may take at least 100 years to be established” This statement does not appear to be supported by your results. A previous paragraph stated that a founder group of 23 reached K within a few decades. It is also unclear whether you are talking about captive or wild or both. Please revise to clarify.

Author Response

Dear Reviewer #1

We would like to submit "Response to Comments and Suggestions for Authors
"

Thank you for addressing many of the last comments and suggestions. This version is improved.

Response: Thank you for your positive comments and suggestions. We revised as follow:

The new objectives statement helps clarify the intentions of the paper, but is a really long run-on sentence. Please divide into 2 or more sentences: e.g., The objectives of this study were to determine the age structure, growth (weight and height), body condition score to developing banteng body metrics and assessments that indicate the ability of the captive breeding program to raise healthy captive individuals. We also projected the number of captive individuals and the age structure into the future to assess the ability of the captive population to indicate continued viability of captive-bred banteng population and the ability of the program to yield animals for reintroduction to ex-situ populations in Thailand and elsewhere.

Response: We change the objective to “The objective of this study was to determine the age structure, growth (weight and height), body condition score to developing banteng body metrics and assessment. We also projected the number of captive individuals and the age structure into the future to assess the ability of the captive population to indicate continued viability of captive-bred banteng population and the ability of the program to yield animals for reintroduction to ex-situ populations in Thailand and elsewhere” in line 80-85.

English edit needed in some places. 

182: Rephrase: the assessment of the three assessor groups were similar, and move to after the second sentence, when the topic changes.

Response: We rephrased as “The body condition scores of banteng increased with age as evaluated by the three assessor groups…” in line 182-183.

238: “If a banteng population is lower than ~20 individuals, a stable population may take at least 100 years to be established” This statement does not appear to be supported by your results. A previous paragraph stated that a founder group of 23 reached K within a few decades. It is also unclear whether you are talking about captive or wild or both. Please revise to clarify.

Response: We changed to “If a captive banteng population is lower than ~20 individuals, to increase the population to an expected stable population for 100 years may take longer than a group of higher than 20 individuals” in line 238-240.

Best regards,

Reviewer 2 Report

The manuscript is significantly improved and I am satisfied with the revisions.  

Author Response

Dear Reviewer

Thank you for your positive comments.

Best regards,

Authors